# Impact of Residual Strains on the Carrier Mobility and Stability of Perovskite Films

**DOI:** 10.3390/nano14151310

**Published:** 2024-08-03

**Authors:** Moulay Ahmed Slimani, Luis Felipe Gerlein, Ricardo Izquierdo, Sylvain G. Cloutier

**Affiliations:** Département de Génie Électrique, École de Technologie Supérieure, 1100 Rue Notre-Dame Ouest, Montréal, QC H3C 1K3, Canada; moulay-ahmed.slimani.1@ens.etsmtl.ca (M.A.S.); luis.felipe@lacime.etsmtl.ca (L.F.G.); sylvaing.cloutier@etsmtl.ca (S.G.C.)

**Keywords:** perovskite, Williamson Hall, GIXRD, CH_3_NH_3_PbI_3−x_Cl_x_, Hall effect, raman

## Abstract

Solution-based inorganic–organic halide perovskites are of great interest to researchers because of their unique optoelectronic properties and easy processing. However, polycrystalline perovskite films often show inhomogeneity due to residual strain induced during the film’s post-processing phase. In turn, these strains can impact both their stability and performance. An exhaustive study of residual strains can provide a better understanding and control of how they affect the performance and stability of perovskite films. In this work, we explore this complex interrelationship between residual strains and electrical properties for methylammonium 
${\mathrm{CH}}_{3}{\mathrm{NH}}_{3}{\mathrm{PbI}}_{3-x}{\mathrm{Cl}}_{x}$
 films using grazing incidence X-ray diffraction (GIXRD). We correlate their resistivity and carrier mobility using the Hall effect. The 
$\sin^{2}\left(\psi \right)$
 technique is used to optimize the annealing parameters for the perovskite films. We also establish that temperature-induced relaxation can yield a significant enhancement of the charge carrier transports in perovskite films. Finally, we also use Raman micro-spectroscopy to assess the degradation of perovskite films as a function of their residual strains.

## 1. Introduction

Perovskite is a promising material with unique optical and electrical properties [1]. However, it can be very sensitive to environmental factors, including humidity, oxygen, and heat [2,3]. These can make the perovskites unstable and limit their integration into commercial devices. Recently, reports have suggested that stability enhancement could be achieved by the release of residual strains [4]. Although residual strains are well known, the first studies on strains in perovskites were reported less than a decade ago [5]. The problems associated with residual strains and their impact on material and device stability remain an active research field [6]. In time, this deeper understanding could translate into a significant improvement in perovskite devices. In this work, we use thermally-induced strain release to explore the origin of residual strains in perovskite films and their impact on their electrical and optical properties, as well as their stability.

### 1.1. Stress Origin in Perovskite Films

Generally, stresses in perovskite films mainly have three origins: intrinsic, mechanical, and thermal [7]. Because of the polycrystalline and heterogeneous nature of perovskite films, residual strains can stem from macroscopic, mesoscopic, or microscopic deformations [8,9]. Residual strains in thin films are usually the consequence of thermal stresses or other external conditions occurring during grain coalescence or growth [4,10]. Indeed, it is largely due to the coefficient of thermal expansion (CTE) and lattice offset between the film and substrate [5,11]. These stresses can modify the structure of the crystal lattice and consequently impact the mobility of charge carriers, the band gap, and the stability of the film [12,13].

### 1.2. Impact on Electrical, Optical Properties and Stability

Recently, perovskite-based solar cells (PSCs) have achieved a certified efficiency of 26.1% [14]. However, instability remains the main factor limiting their large-scale commercialization [15]. Efforts were made to reduce the moisture sensitivity of perovskite materials and improve the chemical stability of the electron and hole transport layers [16,17]. Strains can cause distortion of the crystal structure of perovskite materials [4,18]. Band structure calculations suggest that the band gap increases as the strain changes from compressive to tensile [19]. Small lattice strains in perovskite thin films can have major consequences for carrier recombination, electronic band structure, crystallinity, and crystal phase stability [20,21,22]. The bandgap variations due to strains can result in a change in the absorption spectrum of the perovskite film [5]. The presence of surface defects and grain boundaries in perovskite films also creates trap states, impacting their stability [23]. It is also well known that strains can weaken the bonds and increase the defect density and degradation in perovskite films [12,24,25].

### 1.3. Regulation of Strain in Perovskite Films

Residual strains in perovskite films can affect their physical, optical, and electrical properties and impact their stability [26]. In turn, releasing residual strains in perovskite films could potentially be used to controllably improve their properties or stability. Several techniques can be used to release strains, including (1) regulation of local strains through synthesis, (2) stress release by thermal expansion, or (3) adjusting the lattice offset between film and substrate [5]. It has been established that the intrinsic instability of perovskite films can result from annealing-induced residual strain accelerating their degradation [12,27]. This work will focus exclusively on the temperature-induced strain relaxation of methylammonium lead halide perovskite films in order to improve their optical and electrical properties and increase their stability. A comprehensive study of how residual strain impacts the electrical properties of perovskite films can be conducted using the Williamson-Hall plot, Grazing Incidence X-ray Diffraction (GIXRD), Hall effect measurements, and Raman spectroscopy. The Williamson-Hall plot quantifies strain by analyzing the broadening of X-ray diffraction peaks. GIXRD provides detailed structural information about surface layers, phase analysis, degree of crystallinity, texture, and depth profiling, revealing the strain distribution [28]. Hall effect measurements determine carrier concentration and mobility, showing how strain affects electrical conductivity. Raman spectroscopy detects changes in lattice vibrations due to strain, offering insights into the correlation between lattice dynamics and electronic structure. Together, these techniques elucidate the influence of residual strain on both the structural and electrical properties of perovskite films, guiding their optimization for enhanced performance. Our findings indicate that achieving temperature-induced relaxation requires annealing within the temperature range of 75 to 85 
°C
 to achieve a high carrier mobility, low activation energy, high mean free path, and improved stability. We also find that this optimized annealing process reduces defect density.

## 2. Experimental Section

Glass substrates (25 mm × 25 mm) were cleaned sequentially with acetone, isopropanol, and DI solution for 15 min each. Perovskite layers were prepared with commercial 
${\mathrm{CH}}_{3}{\mathrm{NH}}_{3}{\mathrm{PbI}}_{3-x}{\mathrm{Cl}}_{x}$
 ink from Ossila, Sheffield, UK (used as received). The films were deposited on the substrates by spin-coating at 2000 rpm for 30 
s
 (the ink quantity used per film is 70 µL). The resulting films are annealed on a hot plate for 2 h at different temperatures, ranging from 50 
°C
 to 110 
°C
. The residual strains are characterized using XRD. Hall-effect measurements are used for carrier mobility. Finally, UV-Vis absorption and Raman micro-spectroscopy are used to assess perovskite film degradation. Raman characterization was done using a WITec Alpha 300 confocal Raman microscope (Ulm, Germany). equipped with a continuous-wave 60 mW laser emitting at 532 nm whose power output was attenuated mechanically. XRD was done using a Bruker D8 Advance (Billerica, MA, USA), equipped with a Cu source. SEM imaging was done using a SU8230 from Hitachi (Tokyo, Japan). Optical absorbance was done using a UV-Vis-NIR spectrophotometer by Perkin Elmer (Waltham, MA, USA), model Lambda 750, with an integrating sphere. The Hall effect measurement was done using a Room-Temperature Hall Effect system fabricated by TeachSpin (Buffalo, NY, USA), which uses a permanent magnet that generates fields of about 0.7 T to create the Hall effect. The system is designed to accommodate TeachSpin sample holders, which are printed circuit boards. The perovskite film was deposited on the sample board, which was then mounted on the device chassis and connected to a DC power source, allowing current to flow along the perovskite sample under the effect of the permanent magnetic field, enabling Hall voltage measurement.

## 3. Results and Discussion

In Figure 1, characterization of the films by X-ray diffraction (XRD) confirms the crystallization of a sample in a perovskite structure, with the presence of the standard peaks associated with the (110), (220), and (330) reflection planes, respectively. The peak intensity at 14° suggests that the main orientation of the perovskite film is (110) [29]. XRD is also useful to characterize the residual strains through the lattice parameters. Indeed, the tensile or compressive strains can be respectively associated with the shift of the peaks to lower or higher diffraction angles. Residual strains are often non-uniform and complex to analyze from the peak position alone. Another indicator of strain is the broadening of the XRD peaks compared to the fully relaxed state [19,30].

### 3.1. Williamson Hall Characterization

The Williamson Hall (W-H) analysis, described by Equation (Equation 1) can be used to quantify the strain-induced broadening of the XRD peaks resulting from crystal imperfections [31,32,33], and the structural imperfections such as dislocations, vacancies and stacking defects [34,35]:
(1)
$$\beta_{hkl}\;\cdot \;cos\Theta_{hkl}=k\frac{\lambda }{D}+4\varepsilon \;\cdot \;sin\Theta_{hkl}$$

where 
$\beta_{hkl}$
 is full width at half maximum (FWHM) of the XRD peak, k is the Scherrer constant equal to 0.94, 
λ
 is the X-ray source wavelength, D is the crystallite size, 
$\theta_{hkl}$
 is the peak position (
°
) and 
ϵ
 is the strain. The fundamental difference between W-H’s method and Scherrer’s method is that W-H considers both the crystallite size and microstrains which are often interrelated.

The Williamson-Hall plot in Figure 2a show the 
$\cos\theta_{hkl}$
 vs. 
$4\sin\theta_{hkl}$
 evolution measured from 
${\mathrm{CH}}_{3}{\mathrm{NH}}_{3}{\mathrm{PbI}}_{3-x}{\mathrm{Cl}}_{x}$
 films annealed at 25, 50, 70, 80, 90 and 110 
°C
. The residual strain can be calculated directly from the slope for each curve fitting. In Figure 2b, the strain values calculated reveal an interesting trend. It starts with a relatively high compressive strain at low annealing temperatures, below 40 
°C
. The residual strains decrease as the annealing temperature increases to 70 
°C
. Over 90 
°C
, the residual strains become increasingly tensile. We can conclude from this first test that the annealing temperature can generate three types of strain: compressive, tensile, and a relaxed strain at a temperature around 80 
°C
. Based on our observations, we can expect improved performance from the perovskite film.

It is important to note that a full profile analysis, which involves a comprehensive examination of all aspects of the diffraction data, including peak shapes, positions, and asymmetry factors, is a more robust approach than the Williamson-Hall method presented in this manuscript. The Williamson-Hall method often simplifies certain aspects of the analysis by focusing on the broadening of diffraction peaks to estimate deformations and crystallite sizes. For this reason, we will be combining GIXRD analysis with Williamson-Hall analysis to confirm residual strain trends in perovskite films, validate the results obtained, and overcome the limitations and potential inconsistencies of the Williamson-Hall method.

### 3.2. Gixrd Characterization

Grazing incidence XRD is a very useful technique to characterize the structure of thin films. It can also directly probe the strain distribution in perovskite films [5]. As such, we can use GIXRD to analyze the variations in the in-plane residual strains for the perovskite films annealed at different temperatures. The measurements shown in Figure 3a are performed in the 
ω
-mode [5,36], by fixing the angle 
$\theta_{hkl}$
 and varying the instrument’s tilt angle 
ψ
.

In Figure 3b,c, we analyze the strains in perovskite films annealed at 50, 70, 80, 90, and 110 
°C
 using GIXRD. Figure 3b illustrates the plot of 
2θ
 vs. 
$\sin^{2}(\phi$
), where the slope of the fitted curves yields the strain coefficients. A negative slope suggests tensile strains, while a positive slope suggests compressive strains. Figure 3d–h clearly illustrates the distribution of compressive and tensile strain gradients for annealing temperatures from 50 to 110 °C, respectively. While thermal energy can cause significant strain levels in the crystal lattice, it should be noted that the perovskite crystal structure still remains thermally stable between 50 
°C
 and 110 
°C
 [37]. These GIXRD strain measurements are consistent and strongly support the W-H results presented in Figure 2. In Figure 3, residual strains appear compressive at temperatures below 70 
°C
 and tensile at temperatures above 90 
°C
. We can infer that strain relaxation can be achieved between 75 and 85 
°C
. Based on the W-H, XRD and GIXRD analysis, we concur that 
${\mathrm{CH}}_{3}{\mathrm{NH}}_{3}{\mathrm{PbI}}_{3-x}{\mathrm{Cl}}_{x}$
 has a relaxed cubic structure when annelaed at 80 ± 5 
°C
 (At 80 
°C
 and up, the mixed crystals of perovskite transform from tetragonal to cubic phase), and this relaxed symmetrical structure is favorable to improved electrical properties [37,38].

### 3.3. SEM Characterization

SEM analysis can be used to examine the change in granularity and microstructure caused by the residual strains. Distortions and internal strains can arise from the growth of superstructures and nanostructures, as well as from substitutions with elements of different sizes [6]. Figure 4a–e show SEM micrographs of the perovskite films after annealing at 50, 70, 80, 90, and 110 
°C
, respectively. The grain size evolution indicates that grains appear at 50 
°C
 (Figure 4a). The presence of solvents in the precursor leads to high pinhole densities due to minimal evaporation. Increasing the annealing temperature reduces pinhole density and increases the average grain size [39]. The maximum average size of 795 
nm
 is achieved with 80 
°C
 annealing (Figure 4h). At 90 
°C
 and above, microstructural changes (grain size and surface filing) start to appear (Figure 4d,i). These strain-induced alterations in the perovskite film’s granularity and microstructure could potentially compromise its stability and accelerate its degradation.

### 3.4. Hall Effect Characterization

Carrier mobility can be directly measured using the Hall-effect technique represented in Figure 5a. Here, a transverse magnetic field of 0.7 
T
 is applied to all samples. The applied magnetic field will induce a Lorentz force opposing the movement of charge carriers, separating holes from electrons and creating a potential difference, the Hall voltage (
$V_{H}$
). From this measurement, we can deduce the carrier density (*n*), Hall constant (
$R_{H}$
), charge carrier mobility (
μ
), resistivity (
ρ
), and charge carrier mobility (
μ
) using the following expressions [40,41]:
(2)
$$V_{H}=\frac{B\;\cdot \;I}{q\;\cdot \;n\;\cdot \;d}$$


(3)
$$R_{H}=\frac{1}{q\;\cdot \;n}$$


(4)
$$\mu =\frac{V_{H}\;\cdot \;L}{W\;\cdot \;B\;\cdot \;V_{r}}$$


(5)
$$\rho =\frac{1}{n\;\cdot \;e\;\cdot \;\mu }$$

with *B*: magnetic induction, *I*: injected current, *n*: carrier concentration, *q*: carrier charge, *W*: perovskite sample width, *L*: its length and *d*: its thickness.

In Figure 5b, all samples display a p-type behavior indicated by the positive Hall voltage. This expected result can be explained by the large amounts of Pb, 
${\mathrm{CH}}_{3}{\mathrm{NH}}_{3}I$
, Cl vacancies and I present in the film [42,43]. Due to their lower formation energies, Pb and 
${\mathrm{CH}}_{3}{\mathrm{NH}}_{3}$
 vacancies are known to play a significant role in the p-type behavior of 
${\mathrm{CH}}_{3}{\mathrm{NH}}_{3}{\mathrm{PbI}}_{3-x}{\mathrm{Cl}}_{x}$
 thin films [44,45,46,47]. Figure 5c shows the Hall effect results, illustrating the corresponding change in carrier mobility and resistivity with increasing annealing temperatures. The carrier mobility reaches its highest value of 10 
${cm}^{2}V^{-1}s^{-1}$
 at 80 
°C
, which is consistent with previous reports [48,49]. We already established that 
${\mathrm{CH}}_{3}{\mathrm{NH}}_{3}{\mathrm{PbI}}_{3-x}{\mathrm{Cl}}_{x}$
 films annealed at this temperature are strain-relaxed. Above 75 
°C
 (relaxation zone), the structure adopts cubic symmetry and can achieve high stability since the entropy reduction in the inorganic cage compensates for the high dynamic disorder of the organic cations in methylammonium [38,50]. This cubic phase is also known to offer better electronic properties than the orthorhombic and tetragonal phases for symmetry reasons [51,52,53].

The thermal behavior of Hall mobility can be described by the expression [54]:
(6)
$$\mu_{T}\left(T\right)=\mu_{0}\left(T\right)\;\cdot \;exp\left(\frac{E_{a}}{k_{B}\;\cdot \;T}\right)$$

where 
$\mu_{0}$
 is the exponential prefactor, 
$k_{B}$
 is the Boltzmann constant, and 
$E_{a}$
 is the activation energy. Figure 6a shows 
$ln(\mu_{H})$
 as a function of 1000/T measured between 25 and 110 
°C
. The activation energy 
$E_{a}$
 corresponds to the potential energy barrier height [54], and it can be directly extracted from the slope of the linear fit. From Figure 6a, the activation energy for perovskite films with compressive strains is found to be 400 
meV
, while tensile strain reduces the activation energy to 50 
meV
. These results are also consistent with previous reports [55]. This change in activation energy originates from the slope at approximately 80 
°C
, suggesting significantly reduced activation energy due to film relaxation. It has been previously reported that 
${{\mathrm{CH}}_{3}{\mathrm{NH}}_{3}}^{+}$
 and 
$I^{-}$
 vacancies can easily migrate to a neighboring site due to their low activation energy [56]. This variation of the activation energy with temperature can be directly associated with the film’s residual strains [57].

Using the Hall mobility measurements from Figure 6a, it becomes possible to access the mean carrier path length 
$L_{m}$
 and the mean free time 
$\tau_{m}$
, both represented in Figure 6b, in the 
${\mathrm{CH}}_{3}{\mathrm{NH}}_{3}{\mathrm{PbI}}_{3-x}{\mathrm{Cl}}_{x}$
 thin films using the conventional Drude-Sommerfeld model [45]:
(7)
$$\mu_{T}\left(T\right)=\mu_{0}\left(T\right)\;\cdot \;exp\left(\frac{E_{a}}{k_{B}\;\cdot \;T}\right)$$


(8)
$$L_{m}=\mu_{H}\left(\frac{{\left(3m_{h}^{*}\;\cdot \;k_{B}\;\cdot \;T\right)}^{2}}{e}\right)$$

where 
${m_{h}}^{*}$
 is the effective mass of the hole given by the expression:
(9)
$$E_{a}=\frac{e^{4}\;\cdot \;m_{h}^{*}}{2{\left(2\varepsilon_{0}\;\cdot \;\varepsilon_{r}\;\cdot \;h\right)}^{2}}$$

where 
$\epsilon_{0}$
 is the permittivity of free space, and 
$\epsilon_{r}$
 is the relative dielectric constant used between 5.6 (high frequency) and 25.7 (low frequency) [45]. For the calculation of 
$m_{h}^{*}$
 we assumed 
$\epsilon_{r}$
 = 9 based on the hydrogen model [58]. The values of 
$m_{h}^{*}$
 are estimated at 0.3 
meV
 and 2.4 
meV
 for the tensile and compressive films, respectively [45,59,60], where 
$m_{e}$
 represents the rest mass of the electron.

Results from Figure 6b suggest that films subjected to compressive and tensile strains respectively have lower 
$L_{m}$
 and 
$\tau_{m}$
, respectively. Both parameters peak at 80 
°C
, which corresponds to the relaxation region (strain-free) regime. At this temperature, mixed halide perovskites make the transition from the tetragonal to the cubic phase due to the tilt of the inorganic 
${\mathrm{PbI}}_{6}$
 octahedron and the rotational shift of the organic 
${{\mathrm{CH}}_{3}{\mathrm{NH}}_{3}}^{+}$
, resulting in increased average mobility [53]. Consequently, these results confirm reports that phase shift can change the physical properties of mixed halide perovskites films and positively influence their electrical properties [45].

Figure 7a,b displays the absorption spectrum for samples annealed at 60 
°C
, 80 
°C
, and 110 
°C
 exhibiting compressive, relaxed, and tensile residual strains, respectively. The spectra in Figure 7a show stronger UV absorption for the tensile-strained and relaxed samples. When the residual strains transition from compressive to tensile, the bandgap of the film in Figure 7c show a slight increase from 1.56 eV to 1.57 eV. After four days of storage in an ambient environment, the absorption spectra in Figure 7b is significantly reduced for both tensile and compressive-strained samples, compared to the sample annealed at 80 
°C
. Moreover, as the strained samples become transparent, this degradation also translates in a significant bandgap increase from 1.59 eV to 2.34 eV seen in Figure 7d–f. The low absorption of these samples can be attributed to incomplete crystallization during annealing at 60 
°C
. Poor-quality perovskite films often result from incomplete DMF solvent evaporation. Conversely, high-temperature annealing at 100 
°C
 promotes rapid solvent evaporation, preventing uniform surface filling and poor film quality as shown in Figure 4d.

### 3.5. Raman Characterization

Raman micro-spectroscopy measurements from Figure 8a,b are performed with a very low laser power of 0.5 
mW
 at 532 
nm
 wavelength, using a 10× objective, to avoid any laser-induced damage. All measurements are done under ambient conditions. To perform the Raman measurements, we slowly increase the laser power until the main perovskite peaks are visible. Then the laser is shifted to a new position to record the Raman spectrum of a pristine perovskite. To avoid the laser intensity from impacting the perovskite film’s crystallization, we utilized Equation (Equation 10) to estimate the sample’s temperature based on the intensity ratio between Stokes/Anti-Stokes peaks [61,62] as shown in Figure 8b:
(10)
$$\frac{I_{s}}{I_{as}}={\left(\frac{\vartheta_{0}+\vartheta_{v}}{\vartheta_{0}-\vartheta_{v}}\right)}^{4}e^{\left(\frac{E_{p}}{k_{B}\;\cdot \;T}\right)}$$

where 
$k_{B}$
 is the Boltzmann constant, 
$\nu_{0}$
 is the frequency of the excitation source, and *T* is the sample’s temperature.

The inorganic-organic sublattices of the perovskite have different vibrational frequencies covering the range from 50 to 150 
${cm}^{-1}$
 with phonon energies of 6–11 
meV
 (50–90 
${cm}^{-1}$
) and 11–20 
meV
 (90–150 
${cm}^{-1}$
) [63]. Figure 8c–e display the statistical temperature graphics calculated using Equation (Equation 10), for samples with residual tensile strains, no strains, and compressive strains for a phonon energy of 12 
meV
 at a peak Raman shift of 100 
${cm}^{-1}$
 [63]. The maximum temperature reached by the samples due to laser-induced heating is 324 °K. This value remains lower than the lowest annealing temperatures, ensuring that the 532 
nm
 laser-induced heating does not impact the crystalline structure of the different perovskite samples.

Finally, we can also evaluate the impact of different strain levels on the film degradation using Raman micro-spectroscopy. To do so, measurements are performed every 30 min for 4 days at a sufficiently low excitation power to prevent laser-induced heating. The Raman spectrum in Figure 8a clearly indicates the presence of vibrational peaks at 53, 65, 92, and 100 
${cm}^{-1}$
 with a broader Raman band at 170 
${cm}^{-1}$
, which can be attributed to the Pb-I and Pb-Cl perovskite layers [64]. The sharp peaks between 53 
${cm}^{-1}$
 and 92 
${cm}^{-1}$
 are attributed to the bending and stretching of Pb-I bonds, which are modes of inorganic cages [65]. Meanwhile, the bands at 100 
${cm}^{-1}$
 can be attributed to the vibrations of the organic 
${{\mathrm{CH}}_{3}{\mathrm{NH}}_{3}}^{+}$
 cations [66]. After 24 h, the peak intensity decreases, and no new vibrational bands are observed, implying the structure remained unaltered. It is known that the incorporation of 
$H_{2}O$
 into the crystal lattice is measured by the solvating of 
${\mathrm{MA}}^{+}$
 and the dissolving of the cations [67,68]. The absence of 
${\mathrm{MA}}^{+}$
 can increase defect density and cause a slight displacement of atoms in the crystalline structure, leading to fluctuations in vibrational bands [68,69]. Figure 8a shows a redshift for all bands, suggesting an increased length of the chemical bond. Previous research confirms that the redshift in these bands results from stress exerted by the 
$H_{2}O$
 molecule on the atomic bond related to this vibrational mode and the shift induced by the 
${\mathrm{MA}}^{+}$
 vacancies [68]. However, the observed shift is not consistent, as the penetration of moisture in the film is not uniform due to the heterogeneity of the microstructural morphology, defects, and internal stresses.

In Figure 9a–c, we compare the degradation of the strained films. As expected, results from Figure 9d–f clearly suggest that thermally-relaxed films show lower degradation compared with tensile- or compressively-strained films. Indeed, the average peak intensity decrease for the thermally-relaxed film never exceeds 10%. In contrast, films with residual tensile and compressive strains, respectively, show a decrease in peak intensity of more than 20% and 45%. This is consistent with previous observations that strained films degrade more rapidly due to their crystal structure, which enables easier incorporation of 
$H_{2}O$
 molecules. Surface defects and strain-induced distortions in the crystal lattice weaken the structure, rendering it less resilient to external factors and thus accelerating its degradation [70,71].

## 4. Conclusions

This study shows a clear relationship between temperature-induced strains, charge carrier mobility, and the long-term stability of solution-based perovskite films. Our findings show that these factors are interrelated and must be carefully considered to optimize the performance of perovskite materials. Our findings indicate that achieving temperature-induced relaxation requires annealing within the temperature range of 75 to 85 
°C
 for this specific precursor formulation. As a result, the 
${\mathrm{CH}}_{3}{\mathrm{NH}}_{3}{\mathrm{PbI}}_{3-x}{\mathrm{Cl}}_{x}$
 films annealed at 80 
°C
 demonstrate optimal performance, as shown by higher carrier mobility, lower activation energy, a high mean free path, and improved stability. We find that these annealing temperatures reduce defect density and promote strain relaxation. These results could potentially minimize the impact of residual strain on optoelectronic properties and offer an alternative approach to optimize the performance of perovskite solar cells and other perovskite-based optoelectronic devices.

## Figures and Tables

**Figure 1 nanomaterials-14-01310-f001:**
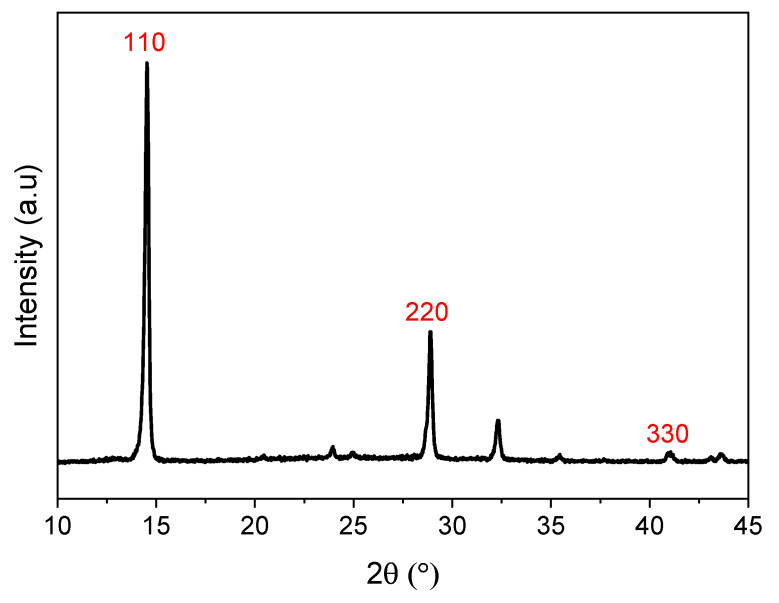
XRD spectra of perovskite film annealed at 80 °C.

**Figure 2 nanomaterials-14-01310-f002:**
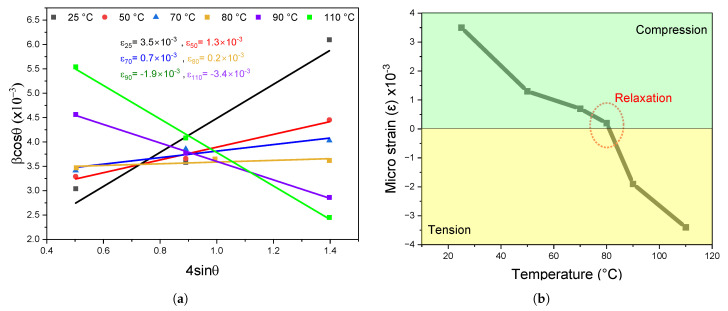
(**a**) Williamson-Hall plot results from which the value of micro-strain was estimated for perovskite films annealed at 50, 70, 80, 90, and 110 °C, (**b**) Residual strain extracted from the slope of the plots in (**a**) using Equation (Equation 1).

**Figure 3 nanomaterials-14-01310-f003:**
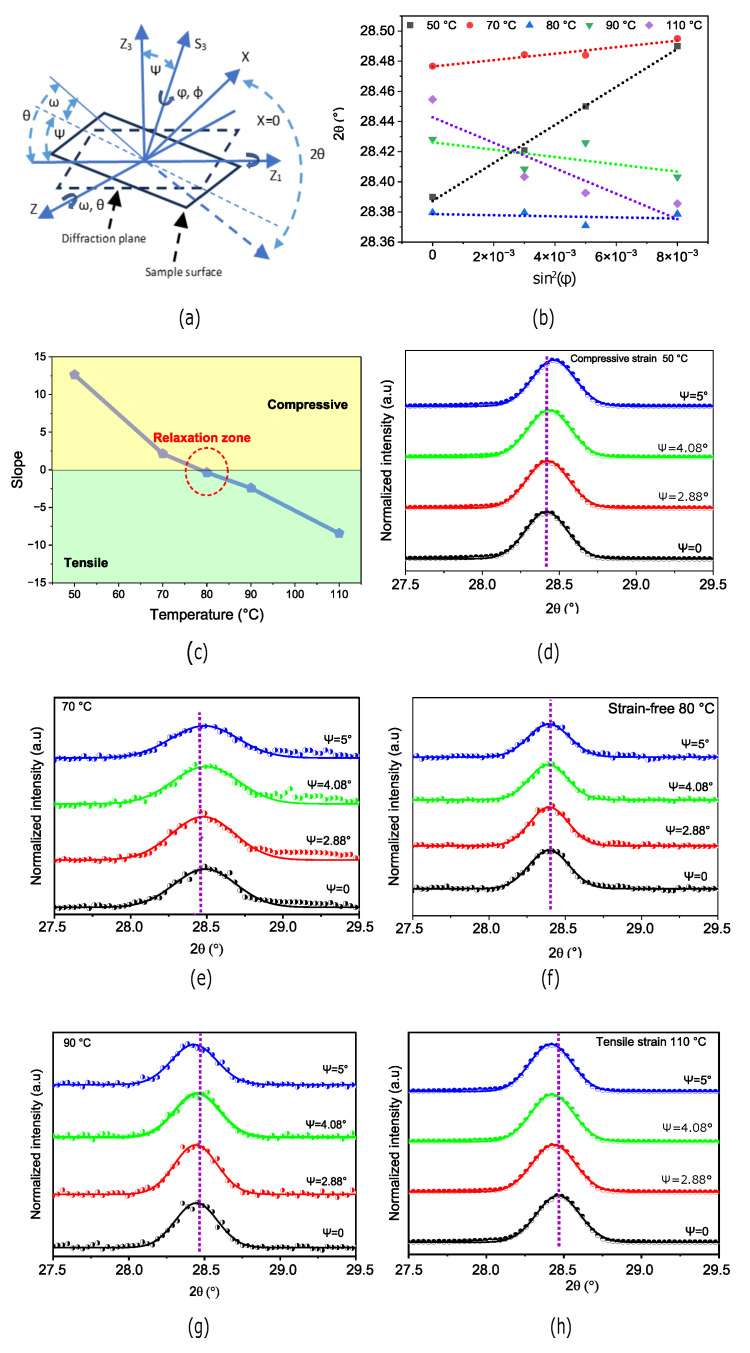
(**a**) Schematic of the strain measurement using GIXRD in the 
ω
-mode, where 
$Z_{3}$
 is normal to the diffraction plane, 
$S_{3}$
 is normal to the sample surface and 
ψ
 is the instrument’s tilt angle. (**b**) 
2θ
 vs. 
$\sin^{2}(\phi$
) from which the value of micro-strain was estimated of the perovskite films annealed at 50, 70, 80, 90 and 110 
°C
. (**c**) Strain coefficients extracted from the GIXRD slope. (**d**,**e**) Compressively-strained films annealed below 80 
°C
. (**f**) strain-free film (**g**,**h**) Tensile-strained films annealed above 80 
°C
.

**Figure 4 nanomaterials-14-01310-f004:**
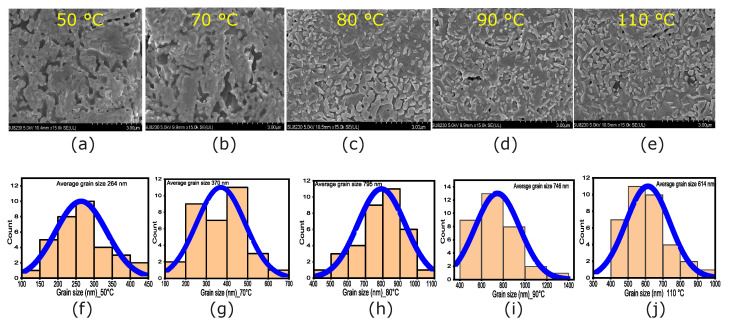
SEM micrographs for the perovskite films annealed at (**a**) 50 
°C
, (**b**) 70 
°C
, (**c**) 80 
°C
, (**d**) 90 
°C
, and (**e**) 110 
°C
. The scale bars correspond to 3 μm. (**f**–**j**) Grain size distributions extracted from the SEM images using ImageJ (V1.53). The blue lines show the Gaussian distribution that describes the data.

**Figure 5 nanomaterials-14-01310-f005:**
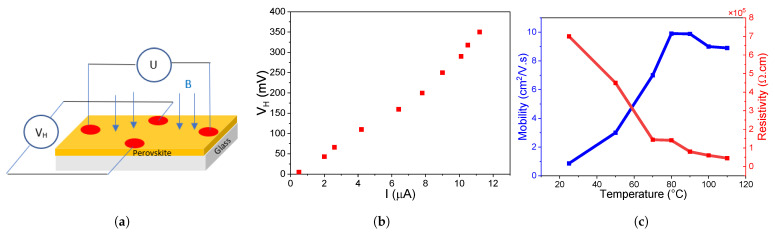
(**a**) Schematic of the Hall effect measurement setup. (**b**) Hall-voltage measured as a function of the current flowing through the device. (**c**) Resistivity and mobility values extracted from the Hall measurements for different annealing temperatures.

**Figure 6 nanomaterials-14-01310-f006:**
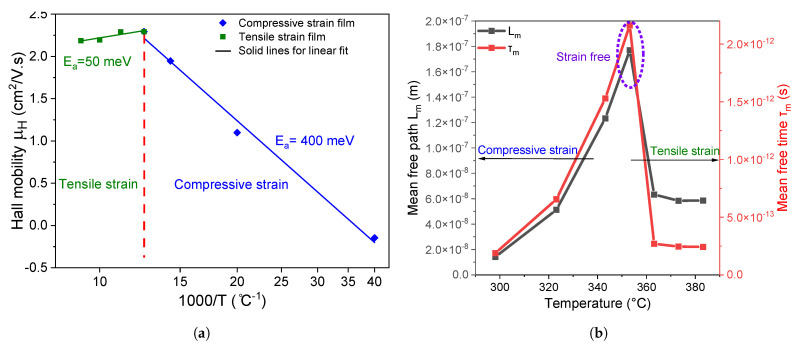
(**a**) Activation energy (
$E_{a}$
) and (**b**) Mean free path (*Lm*) and mean free time (
$\tau_{m}$
) for the 
${\mathrm{CH}}_{3}{\mathrm{NH}}_{3}{\mathrm{PbI}}_{3-x}{\mathrm{Cl}}_{x}$
 films as a function of their annealing temperature.

**Figure 7 nanomaterials-14-01310-f007:**
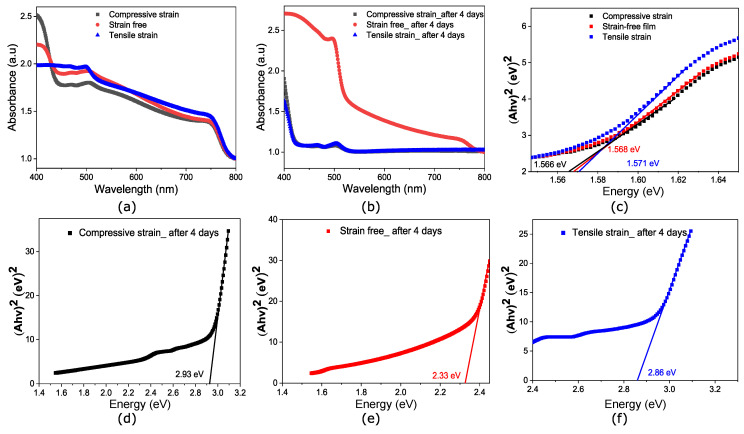
Absorption spectra for perovskite films annealed at 60, 80, and 110 °C. (**a**) Pristine films following fabrication (**b**) After 4 days of degradation in ambient conditions. (**c**) Tauc plot showing the bandgap evolution under temperature-induced strain variation. (**d**–**f**) Tauc plots measured after 4 days in ambient environment.

**Figure 8 nanomaterials-14-01310-f008:**
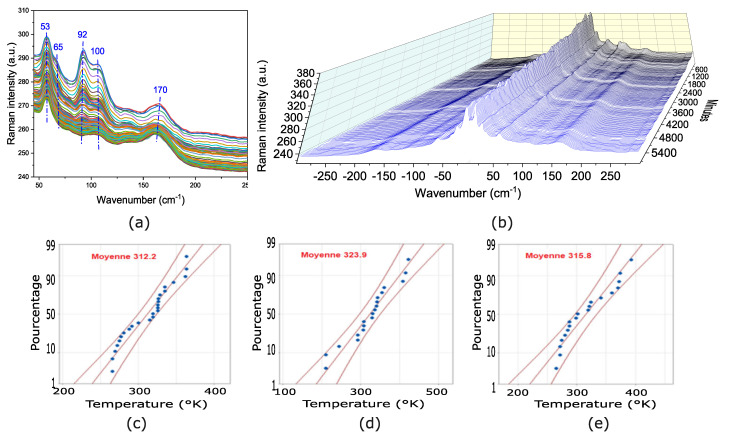
(**a**) Raman peaks shifting of the thermally relaxed sample during 4 days of analysis. Each color corresponds to a spectrum taken after 30 min, (**b**) Stokes/ani-Stokes for samples without strain. Each line corresponds to a spectrum taken after 30 min. Evaluation of the samples based on Stokes/anti-Stokes intensity ratios. Calculated temperature of the (**c**) compressively-strained sample annealed at 60 
°C
, (**d**) thermally-relaxed sample annealed at 80 
°C
, and (**e**) tensile-strained sample annealed at 110 
°C
.

**Figure 9 nanomaterials-14-01310-f009:**
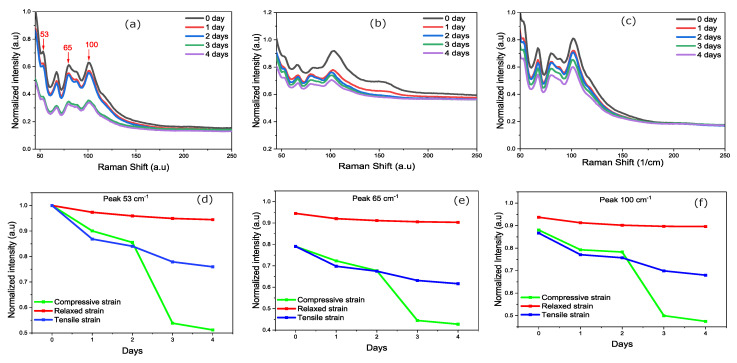
Raman spectra of film degradation at different times from 0 to 4 days. (**a**) With residual compressive strains, (**b**) Thermally relaxed, (**c**) tensile-strained film. (**d**–**f**) Degradation measured from Raman peaks evolutions at 53, 65, and 100 cm^−1^.

## Data Availability

The data that support the findings of this study are available from the corresponding author upon reasonable request.

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
