# Peer review of "Impact of Residual Strains on the Carrier Mobility and Sability of Perovskite Films"

_nanomaterials, 2024, doi:10.3390/nano14151310_

Round 1

Reviewer 1 Report

Comments and Suggestions for Authors

Moulay Ahmed Slimani et al. in the manuscript “Impact of Residual Strains on the Carrier Mobility and Stability of Perovskite Films”, the influence of residual stresses in the perovskite lattice on such important material parameters as carrier mobility and stability was carefully studied. Note that the strains in the crystal were measured in this work by a direct method using such a highly sensitive technique as grazing incident X-ray diffraction (GIXRD). As a result, the authors clearly showed that the stability of perovskite films increases by an order of magnitude or more if both compressive and tensile stresses are eliminated and films with a relaxed lattice are used. At the same time, in such films the charge carrier transport increases significantly. This work is of fundamental importance for the entire scientific field, since it points to what appears to be the main reason for the large scatter of experimental results obtained for thin films of perovskites if the absence of residual stresses is not carefully controlled.

The work can be published in the Nanomaterials journal as it is.

Author Response

Thank you for your time and effort in working on this paper

We thank the reviewer for the extensive criticism and comments that made this manuscript better and more substantial. All the changes that were made to the original manuscript are highlighted in yellow for new inclusions and stroked-out for removals. When necessary, the initial line number in the manuscript is given to facilitate locating the changed text in the original submission. A version with the highlighted changes is included as Non-published material and a final version of the manuscript is uploaded with this response.

Reviewer 2 Report

Comments and Suggestions for Authors

In this paper, the author used several different characterization techniques to prove the interrelationship between temperature-induced strains, charge carrier mobility, and the stability in solution-based perovskite films. These results offer an alternative approach to optimize the performance of perovskite solar cells and other perovskite-based optoelectronic devices. The paper can be published after a major revision. The following comments are suggested to improve the quality of the manuscript.

1.     There are some errors in the manuscript. The author should double check these errors. For example, “However can be very sensitive to environmental factors including humidity, oxygen and heat;” “Perovskite-based solar cells (PSCs) have achieved efficiencies up-to 24.2 % in just a few years, However, their instability remains the main factor limiting their large-scale commercialization;” where βhkl is the fullwidth halfmaximum”; “the annealing temperature can generates three areas of strains”; “Due to lower formation energies, Pb and CH3NH3 vacancies are known to play a significant role in the p-type behavior of CH3NH3PbI3xClx thin films [41,42]. Due to lower formation energies, Pb and CH3NH3 vacancies are known to play a significant role in the p-type behavior of CH3NH3PbI3xClx thin films [42–44].” The author should double check and revise errors through the manuscript.

2.     The figure captions of Figure 2a and Figure 3a are not correctly written, and the content that describe Figure 2a and 3b are also not consentient with Figure 2a and 3b

3.     The author is suggested to provide the SEM image for the film annealed at 80 â—¦C to support the conclusion.

4.     Please provide the photographs comparison of the different films, before and after 4 days of degradation in ambient conditions.

5.     The author mentioned “high average speed” in both the introduction and the conclusion section. However, it was not mentioned in other part. Please provide a detailed explanation about this.

6.     Please also provide the PL evolution curves for the films that were exposed to ambient conditions.

Comments on the Quality of English Language

Need to be improved

Author Response

We thank the reviewer for the extensive criticism and comments that made this manuscript better and more substantial. All the changes that were made to the original manuscript are highlighted in yellow for new inclusions and stroked-out for removals. When necessary, the initial line number in the manuscript is given to facilitate locating the changed text in the original submission. A version with the highlighted changes is included as Non-published material and a final version of the manuscript is uploaded with this response.

  1. There are some errors in the manuscript. The author should double check these errors. For example, “However can be very sensitive to environmental factors including humidity, oxygen and heat;” “Perovskite-based solar cells (PSCs) have achieved efficiencies up-to 24.2 % in just a few years, However, their instability remains the main factor limiting their large-scale commercialization;” where βhkl is the fullwidth halfmaximum”; “the annealing temperature can generates three areas of strains”; “Due to lower formation energies, Pb and CH3NH3 vacancies are known to play a significant role in the p-type behavior of CH3NH3PbI3−xClx thin films [41,42]. Due to lower formation energies, Pb and CH3NH3 vacancies are known to play a significant role in the p-type behavior of CH3NH3PbI3−xClx thin films [42–44].” The author should double check and revise errors through the manuscript.

  • We thank the reviewer for their comments. Here is the list of corrections made to the original manuscript to address each point brought out:
  • In line 16 the sentence “However can be very sensitive to environmental factors including humidity, oxygen and heat;” was changed to: However, it can be very sensitive to environmental factors including humidity, oxygen and heat.

  • In line 38 the sentence: “Perovskite-based solar cells (PSCs) have achieved efficiencies up-to 24.2 % in just a few years, However, their instability remains the main factor limiting their large-scale commercialization;” was changed to: Recently, perovskite-based solar cells (PSCs) have achieved a certified efficiency of 26.1%. However, instability remains the main factor limiting their large-scale commercialization.

  • In line 95 the sentence: “where βhkl is the fullwidth halfmaximum”; was changed to: full width at half maximum (FWHM) of the XRD.
  • In line 107 the sentence: “the annealing temperature can generates three areas of strains”; was changed to: can generate three types of strain.

  • Line 150 was duplicated and it is now fixed: Due to their lower formation energies, Pb and CH3NH3 vacancies are known to play a significant role in the p-type behavior of CH3NH3PbI3−xClx thin films.
  1. The figure captions of Figure 2a and Figure 3a are not correctly written, and the content that describe Figure 2a and 3b are also not consentient with Figure 2a and 3b

We thank the reviewer for bringing this to notice. The caption in Figure 2(a) has been fixed to better reflect the contents of the Figure. For the caption of Figure 3(a), the caption was changed to: Schematic of the strain measurement using GIXRD in the ω-mode, where Z3 is normal to the diffraction plane, S3 is normal to the sample surface and ψ is the instrument’s tilt angle. The caption for 3(b) has been changed to 2θ vs sin2(Ï•) from which the value of micro-strain was estimated of the perovskite films annealed at 50, 70, 80, 90 and 110 °C to better reflect the contents of the figure.

  1. The author is suggested to provide the SEM image for the film annealed at 80 â—¦C to support the conclusion.

As requested by the reviewer, we have added the SEM imaging of the samples annealed at 80°C and it is now Figure 4(c) and the corresponding histogram in Figure 4(h). This image was taken at the time all the characterization was done but was not included in the initial submission.

  1. Please provide the photographs comparison of the different films, before and after 4 days of degradation in ambient conditions.

We conducted all the tests during the winter period, in environmental conditions. The perovskite we used, being a commercial product, is extremely sensitive to humidity. Currently, during summer time, the relative humidity is above 60 %, which is too high for optimal use of perovskite under the same processing conditions (Perovskite datasheet here). The films produced during this period will quickly deteriorate and will not produce consistent results. While we agree with the reviewer that it will be interesting to have these pictures, the absence of them does not hurt or undermine the quality and validity of our results.

  1. The author mentioned “high average speed” in both the introduction and the conclusion section. However, it was not mentioned in other part. Please provide a detailed explanation about this.

            We thank the reviewer for pointing out this omission from our part. We have changed the sentence “high average speed” in lines 69 and 265 to a more appropriate “high mean free path” which is physically accurate.

  1. Please also provide the PL evolution curves for the films that were exposed to ambient conditions.

We conducted all the tests during the winter period, in environmental conditions. The perovskite we used, being a commercial product, is extremely sensitive to humidity. Currently, during summer time, the relative humidity is above 60 %, which is too high for optimal use of perovskite under the same processing conditions (Perovskite datasheet here). The films produced during this period will quickly deteriorate and will not produce consistent results with those presented in the manuscript. While we agree with the reviewer that it will be interesting to add the PL evolution of our samples, we would like to point to the Raman temporal evolution that highlights the degradation over time for our films, and it is a useful tool to describe the degree of crystallinity of the material analyzed.

Reviewer 3 Report

Comments and Suggestions for Authors

The authors studied the impact of temperature-induced strains on the carrier mobility and film stability of perovskites. The optimal annealing temperature was found to be 80 degree Celsius, and the films annealed at this temperatures showed reduced defect density. The work is interesting and can be published after some minor revisions. 

1. A brief but combined summary of the work should be given at the end of introduction.

2. There seems to a confusion on the unit used for temperature throughout the manuscript. In particular, several inconsistencies and units in graph are presented wrongly. recheck this part. 

3. Is it possible to perform the same analysis for inorganic perovskites and how will it be different? 

4. Detailed information on the hall effect sample preparation and measurement conditions, along with references, should be given. 

Comments on the Quality of English Language

The units should be uniform throughout the manuscript. Minor typos must be corrected.

Author Response

We thank the reviewer for the extensive criticism and comments that made this manuscript better and more substantial. All the changes that were made to the original manuscript are highlighted in yellow for new inclusions and stroked-out for removals. When necessary, the initial line number in the manuscript is given to facilitate locating the changed text in the original submission. A version with the highlighted changes is included as Non-published material and a final version of the manuscript is uploaded with this response.

  1. A brief but combined summary of the work should be given at the end of introduction.

We thank the reviewer for pointing out this absence. A brief summary was added at the end of the introduction section.

  1. There seems to a confusion on the unit used for temperature throughout the manuscript. In particular, several inconsistencies and units in graph are presented wrongly. recheck this part. 

We have used Celsius as the only unit of temperature throughout the document.

  1. Is it possible to perform the same analysis for inorganic perovskites and how will it be different? 

Indeed, it is feasible and a fascinating topic. Nevertheless, the study would require greater depth to cover all aspects. Consequently, it could constitute an intriguing new paper that addresses the same study and offers a comparison between the effect of residual strain in completely inorganic perovskite films and organic-inorganic perovskite presented in this study.

  1. Detailed information on the hall effect sample preparation and measurement conditions, along with references, should be given.

We have detailed the experimental part of the Hall effect, and also added the necessary equations and references in Hall effect characterization section

Reviewer 4 Report

Comments and Suggestions for Authors

The manuscript presents the study of complex interrelationship between residual strains and electrical properties for methylammonium CH3NH3PbI(3−x)Clx perovskite films.  The samples were characterized by grazing incidence X-ray diffraction,  SEM, UV-Vis and Raman spectroscopy. The resistivity and carrier mobility were characterized using Hall effect.   It was established that temperature-induced relaxation can yield significant enhancement of the charge carrier transports in perovskite films. The results of the study can be utilized to minimize the impact of residual strain on optoelectronic properties and pave the way to alternative approach for perovskite performance optimization.

     The paper is well-structured and relevant for the field; the experimental data seem to be reliable. The cited references are relevant.  The figures and images are appropriate; they properly illustrate the data. The conclusions are consistent with the arguments presented.

The following principle points should be mentioned concerning the manuscript content and data presentation:  

1 The authors use «grazing incident XRD» and «grazing incidence XRD» . Please, correct.

2         Equation (1):  cos(theta(hkl)) and sin(theta) or sin(theta(hkl))?

3         The authors use K and °C (Figs. 5,6, for ex.). It is better to use either K or °C.

4         Experimental section is very short. More information is needed on the physical dimensions of the samples. Details on the equipment and methods used for XRD, Hall effect measurements, UV-Vis absorption and Raman micro-spectroscopy should be added. SEM is used but not mentioned in “Experimental section"

5         The Gaussian approximation of the grain size distributions (fig.4) is questionable, especially for cases (e, f).

6         It should be noted that Williamson Hall characterization gives contradictory results in some cases, so a full profile analysis is preferable.

Author Response

We thank the reviewer for the extensive criticism and comments that made this manuscript better and more substantial. All the changes that were made to the original manuscript are highlighted in yellow for new inclusions and stroked-out for removals. When necessary, the initial line number in the manuscript is given to facilitate locating the changed text in the original submission. A version with the highlighted changes is included as Non-published material and a final version of the manuscript is uploaded with this response.

1 The authors use «grazing incident XRD» and «grazing incidence XRD» . Please, correct.

      In line 8 «grazing incident XRD» was changed to grazing incidence XRD

2         Equation (1):  cos(theta(hkl)) and sin(theta) or sin(theta(hkl))?

Corrected

3         The authors use K and °C (Figs. 5,6, for ex.). It is better to use either K or °C.

We have used celcius as the temperature unit throughout the document lines

4         Experimental section is very short. More information is needed on the physical dimensions of the samples. Details on the equipment and methods used for XRD, Hall effect measurements, UV-Vis absorption and Raman micro-spectroscopy should be added. SEM is used but not mentioned in “Experimental section"

More details of experimental tests are added in the Experimental section.

5         The Gaussian approximation of the grain size distributions (fig.4) is questionable, especially for cases (e, f).

Indeed, it is harder to quantify the grain size distribution in the samples annealed at 50°C and 70°C as the crystalline structure has just started to emerge. As such, it was harder to select the right range of sizes to yield an appropriate gaussian distribution. However, after rerunning the particle detection algorithm with better size parameters, the resulting histogram showed a gaussian size distribution. Figures 4(e,f) have been replaced and now are Figures 4(f,g) after the inclusion of the SEM and grain size histogram for 80°C.

6         It should be noted that Williamson Hall characterization gives contradictory results in some cases, so a full profile analysis is preferable.

We fully agree that a complete profile analysis, which involves a meticulous examination of all aspects of diffraction data, including peak shapes and positions as well as asymmetry factors, is more comprehensive than the Williamson-Hall method presented in this manuscript. Indeed, it often simplifies certain aspects of the analysis by focusing on the broadening of diffraction peaks to estimate strains and crystallite sizes. However, in our situation, we have used both the Williamson-Hall and GIXRD techniques to confirm, first, the trends of residual strain in perovskite films and second, to validate the results obtained and to overcome the limitations and possible inconsistencies of the Williamson-Hall method.

We've added a paragraph at the end of the Williamson Hall characterization section, explaining the shortcomings of this technique and how the combination of the two complement each other

Round 2

Reviewer 2 Report

Comments and Suggestions for Authors

Accepted

Comments on the Quality of English Language

Can be improved